# Assessment of Nursing Care and Teaching: A Qualitative Approach

**DOI:** 10.3390/ijerph16152774

**Published:** 2019-08-03

**Authors:** Jose Manuel Martínez-Linares, Rocío Martínez-Yébenes, Francisco Antonio Andújar-Afán, Olga María López-Entrambasaguas

**Affiliations:** Department of Nursing, Universidad de Jaén, 23071 Jaén, Spain

**Keywords:** evidence-based nursing, nursing education research, nursing staff, nursing students, qualitative research, theory-practice gap

## Abstract

Background: The reform of the Spanish higher education to adapt to the European Higher Education Area involves, among other issues, the students’ participation in the curriculum assessment. The aim is to understand the insights of both nursing professionals and current undergraduate students of nursing on the connection between the knowledge acquired throughout the degree and the professional healthcare practice. Methods: An exploratory, descriptive qualitative study was carried out at a Spanish University. By convenience sampling, twelve nursing professionals and twelve 4th-year students of the Degree in Nursing were included. In two phases, twelve semi-structured interviews and two focus groups were conducted in order to triangulate data. A thematic analysis of data was carried out, later to be coded by two researchers. Results: Two main themes were identified: Evidence-based nursing vs. experience-based nursing, and a theory-practice gap. The topics that were specifically highlighted were the coexistence between professionals who work according to evidence and those whose work is based on experience, and the lack of connection between the training received during the degree and actual healthcare practice. Conclusion: Nursing care work varies in terms of the implementation of evidence-based care. Nursing training is perceived as being affected by a theory-practice gap. To achieve the linking between nursing theory and practice, a great effort on stakeholders would be needed.

## 1. Introduction

The Spanish higher education system implemented a process of convergence with the European Higher Education Area (EHEA), which started with the underwriting of the Bologna Declaration [1] in 1999 and which would have been completed by 2010.

The contribution of universities to the Bologna Declaration was reflected in the Tuning Project [2], which focused on quality throughout curricular design, development and implementation. The Bologna Declaration unified nursing curricula and facilitated the validation of qualifications and the mobility of nursing professionals throughout the European Union. To this end, a method for quality improvement was developed to encompass all elements of the learning process. All the countries participating in the Bologna Declaration joined this project, according to which assessment has to be the key tool to verify the success of a certain degree’s curriculum. In order to do so, it is necessary to assess whether graduates of a certain degree achieved the goals set during their studies in terms of competences. University students must participate in the quality improvement of university degrees taught in the EHEA, as indicated under the guidebook elaborated for such a purpose in 2015 [3].

In this constantly changing scenario, new challenges were also posed for teaching professionals. In order to improve the educational model based on professional skills, and having in mind the constant and worldwide problem which is the theory-practice gap [4,5], the design of the curriculum of a discipline such as nursing should attempt to answer questions such as “what knowledge is valid?”, “what should students learn?”, “how should they acquire knowledge, skills, and attitudes?”, and “how should the achievement of competences be assessed and their mastery ensured?” [6].

Evidence-based nursing (EBN) should be the path to the professional practice and may be part of the solutions to the theory-practice gap [7]. Evidence-based practice should become the standard of clinical care [8], changing the traditional “experience-based nursing” model, thus renewing the provision of services and guiding care to effectiveness, efficiency and patient safety [9,10]. However, EBN practice is not always adhered to, which foments the theory-practice gap [11]. Universities teach what nursing is, what are its competences and how to apply professional care. This message should be in accordance with students’ experiences during their clinical practices, and the theory-practice gap might be minimized. In order to achieve this, lecturers and professors would have to make sure that what is taught in class reflects the reality of practice. In order to explore how students and newly-qualified nurses perceive their academic and clinical training, an investigation was designed and developed. One of the aims of the study is reported in this paper.

The aim is to understand the perception of both nursing professionals and current undergraduate students of nursing on the connection between the knowledge acquired throughout undergraduate studies and the professional healthcare practice.

## 2. Materials and Methods

This paper is part of a larger project on the acquisition of nursing skills and the improvement of the academic curricula. The methodology is common to all papers, and the breadth of the results is such that it cannot be included in a single paper. The descriptive qualitative method used is presented following the criteria included in the consolidated criteria for reporting qualitative research format (COREQ) [12]. This method aims to provide a true and faithful account of the reality of the results, by setting use and relationship between categories. It is appropriate for exploring unknown areas of knowledge.

### 2.1. Research Team and Reflexivity

Both interviews and focus groups were conducted by all the authors of the team. O.M.L.E. is RN and PhD, R.M.Y. is RN and RM, F.A.A.A. is BsC, and J.M.M.L. is R.N, R.M. and PhD. At the time of the study, O.M.L.E. was a female lecturer and J.M.M.L. a male lecturer; R.M.Y. was a female associate researcher and F.A.A.A. a male associate researcher. All of them had previous training in qualitative research, and O.M.L.E. had previous experience in qualitative research.

Prior to the study, none of the research team members had had any contact with the people participating in the study. They had only been previously informed of the development of the study, its objectives, and the way to participate in order to obtain voluntary participation.

### 2.2. Study Design

An exploratory, descriptive qualitative study was carried out, since the subject addressed is complex and includes concepts and aspects that cannot be measured [13]. It also allows the research team to reveal the subjective perception and opinion of the participants [14].

A convenience sampling was conducted among nursing professionals who had studied at this university and current undergraduate 4th-year students of the same university. Potential nurses (306 graduated nurses in the last three years) were sent an email that included a brief description of the project, the inclusion criteria, and information on data collection and confidentiality. Twenty of them were willing to participate.

The twelve nursing professionals who finally took part in the study met the following inclusion criteria: (1) 3–12 months’ work experience as healthcare professionals in Spain; (2) the completion of their higher education during the years 2017, 2016, or 2015. Regarding 4th-year students, the criterion was to have taken all of their undergraduate training at this university. With these criteria, people who took part in the study had the necessary professional experience and recent training experience to provide data that responded to the research objectives. 

### 2.3. Data Collection

The data collection was carried out in two phases:

Phase 1: Between March and April 2018, 12 semi-structured personal interviews were carried out with registered nurses. Nine of them were developed in an authorized stay at the university, and three of them were conducted via Skype (Microsoft©, Redmond, WA, USA) as the subjects were not located in the province. The interviews lasted between 45 and 90 min. With such a number of interviews, data saturation was achieved [15], as no data other than that already collected was provided, since the analysis categories were saturated. No one else was present at the interviews. The sociodemographic data of the participants are shown in Table 1.

Phase 2: After the analysis of the phase 1 interviews, the decision was made to recruit 4th-year students in order to triangulate some results obtained regarding the theoretical and clinical training they received. The contribution envisaged from the students was in terms of knowing if some of the issues raised in the interviews were still happening at present. The triangulation of data was intended to achieve the reliability of the data obtained in personal interviews through a different source of information. The recruitment process was the same as in phase 1, and sixty-seven students were invited to participate. Twelve of them agreed and configured the total number of participants, who were organized into two focus groups with six participants each in September 2018 (Table 1).

The focus groups were organized by two 4th-year students, who were collaborating in the research project. When asked about the quality of the teaching received, none of the groups was conducted by principal or collaborating researchers, with the aim of minimizing information bias between the students and the faculty who had given them both theoretical and practical lectures. Collaborating students were instructed on the development of this type of data collection technique beforehand. Both focus groups were interviewed for 60–90 min, and the questions were about their theoretical and practical training. No one else participated in the development of the focus groups. 

The interview questions were drafted ad hoc after a previous review of the literature. This literature included the current legislation on competences and qualitative/quantitative studies, which addressed nursing competences, EBP and the theory-practice gap. The interview guide had three thematic sections: clinical training, theoretical training, and working situation (Table 2). A first version of the interview was carried out with former students of this university to determine the relevance and importance of the questions. The questions were modified according to their contributions.

Each interview and focus group were conducted only once and were both recorded in audio files for later transcription and analysis. The transcription was not returned to the participants for correction. Field notes were also taken in both cases and were incorporated into the data analysis. During the interviews and focus groups, elements of nonverbal communication that provided emphasis and clarified data were thereby collected.

### 2.4. Data Analysis

The data analysis was carried out by two researchers, who jointly presented the important themes for analysis [16]. Two researchers analyzed the transcriptions, each of them independently, by following the six-phase method of thematic analysis with scientific precision (familiarizing themselves with the data, generating initial categories or codes, searching for themes, reviewing themes, defining and naming themes, and producing the final report) by Braun and Clarke [17]. The researchers shared their data after the second phase to ensure use of the same coding. A second meeting took place to define the coding of the resulting themes. The codes used to identify the subthemes and themes are shown in Table 3 below:

Program Atlas.ti version 7 for Windows was used in the analysis process in order to easily manage the data: organizing codes, designing code families and creating networks with emerged themes.

### 2.5. Ethical Considerations

The study was carried out by following the ethical principles of the Declaration of Helsinki. The personal data were processed in accordance with Regulation (EU) 2016/679 of the European Parliament and of the Council of 27 April 2016 on the protection of natural persons with regard to the processing of personal data and on the free movement of such data, and repealing Directive 95/46/EC.

Each participant of face-to-face interviews or focus groups was asked to complete the corresponding informed consent, and had the option to leave at any point. The study was carried out after obtaining the approval of the ethics committee of this university. The institutional permission was granted by the Faculty of Health Sciences. No other approval or permission was needed. The ethical principles of anonymity, confidentiality and informed consent were upheld throughout the study. 

## 3. Results

In this paper, two of the main themes that emerged from the qualitative analysis of this large investigation are reported: “Evidence-based nursing vs. experience-based nursing” and “Theory-practice gap”, with several subthemes (Figure 1). The themes concerning different aims of the study are not reported here.

### 3.1. EBN vs. Experience-Based Nursing

Both clinical practice guidelines and nursing interventions with scientific evidence should be the frame of reference for nursing practice. However, the answers to the question asked to both nursing professionals and current undergraduate students about their clinical rotating internship: “*Did you learn and have the chance to perform care and evidence-based nursing techniques?*”, can be summarized as: “*It largely depends on the department and the professional*”. Interviewees felt that this way of working may be more thoroughly developed in special services such as emergencies, intensive care units or operating rooms, while in inpatient or primary care units, it is almost non-existent:
“In the operating room, for example, I did. Clinical practice guidelines were available whenever I wanted to read them, there was no problem. However, in other units, it is complicated, especially in primary care. In primary care, there is just no guidance practically.”(E12)

In some cases, the failure to apply EBN care as undergraduate students makes them consider this deficiency as a weakness of their own care work when they become professionals. The lack of evidence-based nursing placements during the student practicum causes them not to be performed as professionals either:
“One of my biggest weaknesses in working as a nurse is not being able to explain why some things are done, grounding certain things… I mean… evidence-based nursing practice. When it comes to protocols, there were a lot of them, but none of them were followed.”(E2)

#### Continuing Nursing Education Valued as a Positive Attitude

This subtheme focuses on the coexistence of professionals who are “up-to-date” and complete some form of continuing education, along with others who are not. Continuing education for nursing professionals has never been a mandatory requirement in Spain; however, interviewed professionals reported that young and older nurses work differently. They point out that younger ones carry out continuous training and retraining activities, while older nurses do not:
“If the workers are recently graduated or just young, they are more likely to have been trained in information searching, papers, etc.[…] And then you find people who, although there are protocols of action or clinical practice guidelines, they just don’t follow them because, according to them, it’s how they’re used to working, they do it that way and they think it’s better.”(E5)

This perception is also shared by the interviewed students, for whom, in turn, variations in care practice generate an uncertainty and insecurity about the most correct way to apply care. In order to avoid this, it would be necessary to unify the clinical practice to promote learning in evidence-based nursing care during the training of future nursing professionals. In addition, by applying social learning theory, health settings become the ideal places to learn, by observing and applying what has been learned:
“I think so, there are many contradictions […] and here they are telling you one thing and there you come across another. Then, you get confused, because you don’t know what’s well done and what’s not.”(GF2-P11)

The main highlighted difference was that retrained professionals practice evidence-based nursing more, while non-retrained professionals continue to rely on their own experience to work, even refusing to change.

### 3.2. Theory-Practice Gap

Through the participants’ answers, it is possible to infer the existing gap between the theoretical training of the degree and the healthcare practice. This perception was shown through several subthemes:

#### 3.2.1. The Degree Is Not Oriented to Actual Nursing Practice

Despite the need for consistency/correspondence between what is learned in lectures and what is done in clinical settings, both professionals and students brought to light the lack of connection existing, in their opinion, between the theoretical contents of some courses and the healthcare work:
“Then, reality is other and, OK, they explain a theory to you, but then, when you get to the practice, everything is totally different, so…”(E5)
“That’s fine, but then there are also services […] whose way to work is unknown to me, I just don’t know what I’m going to find there, and there are others that I do know, […] and I think the snag is in the course itself, if the course approaches disorders in a certain way, that is finally the idea that you build in your head…”(GF2-P10)

In this sense, the importance of theoretical contents was highlighted as one of the reasons for this. They consider that there are some essential contents that should be highlighted in training, but instead they are taught others that are not so relevant:
“I believe that the global vision of what a nursing degree is, which is to train general nurses, and not researchers or specialist nurses, has been somewhat lost. […] Because they are very basic things, but things that cannot be taught only in one or two short placements, but in every course.”(GF1-P1)

#### 3.2.2. Academic Training Is Not Oriented to Actual Practice

Another aspect which interviewees highlight is the theory-practice gap in the contents of some subjects within the nursing course. The learning of some techniques that nursing professionals do not actually perform or only do in exceptional cases was pointed out, to the detriment of other techniques that are part of the daily work, but that are not given enough importance:
“But, for example, they teach us intubation… We don’t do intubation; it’s good to know in case that, by any chance, it must be done because there is no one else to do it. But there are also many techniques that you do in the hospital and which you haven’t been taught in university, techniques you eventually learn in the placements.”(E7)
“I want to point out that, we work on a lot of topics during the lectures, topics which then don’t fit in so well with actual practice. The lectures then should be more focused on how to use some devices in order to prevent those typical situations in which we say: I’ve never seen this and I don’t know how to do it.”(GF1-P6)

One of the reasons that are argued for and which may somehow be provoking this, is the fact that some courses have a deficient teaching load that should be extended, to the detriment of other courses whose credit load should be reduced:
“It’s true that the problem is that there are courses that are six-monthly, or perhaps four-monthly, and maybe they should be annual.”(E5)
“I understand that there are courses that have to be there, because they have to, but these don’t contribute much, and they are given a lot of importance. There are very important courses that have lost importance […]”(E1)

In this respect, the teaching staff is another reason why there is a theory-practice gap in the training of future nursing professionals, due to the incompatibilities to which they are subjected due to their work as full-time teachers. From the information provided by students and professionals, it could be deduced that the associate teaching staff generates the theory-practice connection in a more accurate way than the staff focused exclusively on university teaching does. It is interesting to see how the good work carried out by the associated teaching staff, who provide the vision of actual nursing practice, is highlighted:
“Yes, indeed. Nurses who both work and teach transmit knowledge that is much more adapted to reality, you don’t feel that they have the head in the clouds.”(GF1-P5)
“In addition, you can see that there are teachers who haven’t worked in hospitals for many years and you say “that’s impossible…!” […] And you realize it’s impossible. And you make comments like: Yes, that would happen in the ‘90s. And you are like… The difference lies in experience, uh? But… tell me that as an anecdote, it’s not done anymore. […] And besides, some courses are quite old-fashioned, so you say: This was done in the past… but not anymore.”(E3)

#### 3.2.3. Working Activity Lacks Care Plans

The nursing staff and undergraduate students argued that the nursing care process as a working method was not followed rigorously. Although it is usually made clear that it is the best method for applying care, it is not implemented for some reasons that include the shortage of time. For the students, this means spending time learning what is not really performed, while leaving aside other contents and techniques more usual in healthcare work:
“Yes, with regard to the Nursing Care Plan (NCP) that we studied… here you can study them, and I think they’re useful, but… I think they can’t be implemented because it takes a lot of time to carry them out too.”(E3)
“I want to point out that we really worked on the NCP in lectures, which then don’t fit in so well with actual practice. Yes, they have to teach you how to make an NCP, but here in lectures you should be more focused on other topics, because then you start working […] and you say: I’ve never seen this in my life and I do not know how to do it.”(GF1-P6)

#### 3.2.4. Holistic View and Humanization of Care Are Questioned

Finally, participants stated that, when healthcare work begins, the holistic vision of the person is lost, as dehumanization of care appears, even from the beginning of clinical placements. Some reasons for this are the work overload and lack of time to talk to the patient, the greater importance of the performance of increasingly complex techniques, as well as the failure to carry out care plans that take into account not only the physical dimension of the person, but also the emotional and spiritual dimensions:
“It’s the nursing concept I like, but not the one I see, because… it’s true that at university the subject of humanization often arises, and we deal with people and humanize and humanize and humanize and humanize… And that’s from the day you start until the day you leave. And I think it’s very important. But it’s not what you see at the hospital. […] “Patient from room No. X” or… “The man in that room”. People don’t seem to have names.”(E3)

On the other hand, the current undergraduate students’ perception is that, as in the case of the application of evidence-based nursing, it also depends on the professional and the service. This is another element of variability that does not promote the learning of professional future of nursing:
“I think the same, it also depends on the nurse. However, in one service, for example, what caught my attention was that the nurse introduced me to all the patients one by one on the first day, and that was interesting.”(GF2-P10)

## 4. Discussion

In general terms, participants were critical of health care activity and training work carried out by nursing staff. The insights that they provided are of great value and, at the same time, necessary for improving both activities.

According to them, the application of EBN varies depending on the unit and is not a widespread and generalized practice at present. There is a lack of unanimity in the nurses’ way of working in the health system. According to Fleming [18], the process of applying an evidence-based practice [19] is made up of five stages (determining the information needed, searching for bibliographic data to get to know the results of the research carried out in this regard, subjecting those results to criticism, using any evidence available in the care plan, and evaluating the progress of execution in the care plan). Furthermore, there are some difficulties identified by Martínez [20] when it comes to applying evidence in the nursing profession, including an insufficient autonomy in modifying the care activities derived from the research, the lack of time for incorporating new ideas, the lack of support provided by administrations for practical implementations, the lack of credibility on the nurses’ part toward the results obtained in research, the nurses’ lack of mastery in statistical analysis, or their inability to assess the research quality.

During this process, university centers contribute by training future nursing professionals in research methodology, evidence-based practices, critical thinking and sensible judgement, or autonomy at work, among other aspects. However, this offered training is not reflected in the actual healthcare practice, and this leads to confusion among the students, who consider that training focused on actual labor is of greater value. In addition, in our context, nurses dedicated to research are neither visible nor usual.

Studies conducted on the implementation and use of evidence-based practice by nurses show contradictory results. Thus, Dalheim et al. [21] concluded that evidence-based literature searches were more commonly conducted by older and more experienced professionals. However, a few years earlier, Milner et al. [22] obtained contradictory findings on the thesis that nurses with a longer professional experience based their care work on their own experience and conducted fewer searches. This would be in line with the participants’ opinion and would lead to the dichotomy between experience-based nursing and EBN. It should also be borne in mind that, in Spain, nursing studies underwent an important change when they were integrated into the EHEA in 2009 [23], which involved a demand for research methodology training and EBN. That is why there are currently many active healthcare professionals who did not receive any training in those subjects when they were students.

The implementation of the scientific method in the nursing care practice is fulfilled through the NCP, since it makes it possible to provide care in a rational, systematic and continuous way, based on scientific evidence. Although there are no conclusive studies on the actual use of NCPs as a working method in Spain, some studies reveal that the use of this standardized language is a hindrance to nurses. For them, this tool is too confusing, not very agile, not practical at all, with a time-consuming complicated language [24]. Nurses also find a disparity between the theory of taxonomy and its application in clinical practice [25], or they perceive it as an abstract and universal language, disconnected from daily practices and incapable of transmitting the specific features of each person to whom it is applied [26]. In this way, the deficient use of NCPs and the resistance to changing the working methodology based on one’s own experience is evident, despite this being a legal requirement [27]. Santos [28] concludes that theoretical models are poorly implemented in healthcare practice and that little use is made of nursing research papers, especially in journals with higher impact factors. This leads to unjustified variations in clinical practice, which could be potentially harmful for patients and result in poor quality health care for the population [29]. The mere fact of being health professionals constitutes one of the factors that cause this variability, due to demographic, professional and training characteristics [30]. However, studies that reveal the magnitude and importance of the variability in clinical nurse practice, and specific explanatory theories of nurse reality are required [31].

The second major theme was the existing theory-practice gap in undergraduate education. This theme began to be discussed in the scientific literature [32,33,34] in the late 20th century. In general terms, the theory-practice gap can be defined as the discrepancy between the training that students acquire through the lectures and their experience in clinical settings [35].

Rolfe [36] holds that this gap is due to existing outdated theoretical concepts and a misconception about the relationship between theory and practice. Other authors point out that the outdated concepts originated in the lack of integration of research results into clinical practice [37,38,39,40,41]. Therefore, the lack of integration of EBN into the usual clinical practice seems to also be a part of this theory-practice gap. Bennett et al. [42] have recently shown that nursing professionals were required to master skills which had not been taught during their training. Likewise, another reason held in one study was the lack of adaptation between the theoretical and practical contents within some courses [43].

Such findings coincide with the results of this study, which show the lack of connection between the lectures and the clinical placements. In this way, students highlight certain differences between the ways of teaching theoretical contents and the practical ones, while professionals refer to a differentiation between “what the professional should be” according to nursing models and “what the professional is” due to actual care practice. Furthermore, as Arreciado and Isla [44] conclude, clinical placements are a key element for providing an exceptional opportunity to experience professional reality and to be able to contrast it with the reality transmitted in theoretical settings. Moreover, clinical settings allow students to recognize the variety of nursing models that exist, and therefore they offer the possibility for students to choose the one with which they most identify.

The possible solutions suggested to solve such a gap included the use of problem-based learning [32], having lecturers carry out clinical placements and become more updated at work [45] and having teaching and care personnel swap functions regularly [46], or having students be trained and prepared for the actual working situation [47]. In addition, Riksaasen [48] states that “helping students develop reflective skills and strengthening the theoretical parts of the nursing education programme, might be beneficial in promoting coherence between theory and practice in initial nursing education”. Implementing these proposals would result in benefiting from more competent nursing professionals, and to providing quality care to the population. Meanwhile, conclusive evidence is needed before a strategy aiming to better integrate theory into the practice of nursing can be developed [35]. Lecturers are fundamental in achieving these improvements.

In this regard, Medina [49] argues that the cause of this lack of connection is the unequal relationship between theory and practice in academic curricula, which put more emphasis on theory and consider practice to be a mere application of what is learned in the lectures. Thus, the academic curriculum is perceived as poorly oriented toward professional practice. It should also be borne in mind that the adaptation of curricula to the EHEA led to a shift from an eminently practical training to the broadening of the theoretical content to substantiate such a practice. The discipline of nursing had to be built from theoretical knowledge to later become a profession [50].

Professional experience is neither a requirement nor even virtually considered to be a merit when working as a lecturer in the nursing degree, so there are lecturers with little or no experience in healthcare. This makes their vision of actual healthcare rather limited to their experience during placements. This lack of previous professional care experience can also contribute to such a curricular disconnection between theory and practice, as was also argued in the study of Saifan et al. [43].

For the people who are taking part in this study, the vision of the actual nursing practice is provided by the associated teaching staff, who carry out their healthcare and teaching activities simultaneously and are highly valued for this. One of the solutions proposed to integrate practice into theory was the collaboration between healthcare professionals and researchers [51]. 

Finally, another reason behind the theory-practice gap was the loss of the holistic vision of the person and the dehumanization of care that occurs when every nurse starts their professional activity. However, this opinion was nuanced by current undergraduate students, who argued that this does not apply to all nurses. The American Holistic Nurses Association defines holistic nursing as “all nursing practice that considers the whole person as its goal” [52]. All nursing models and theories contained a holistic view of care, but it was Martha Rogers in 1980 who made it more explicit by focusing on the “unitary human being” [53]. Both Hildegard Peplau [54] and Jean Watson [55] promoted the humanization of care by means of their theories.

The dehumanization of care is a multifactorial phenomenon in which the social fabric, health structure, academic training, technical hegemony, patients themselves, and technological means come into play [56]. However, to be optimistic, our participants do not associate dehumanization with the academic training they received.

### Limitations

The submitted results have been extracted from a limited number of subjects, who had received their entire theoretical training and virtually all of their practical training in the university where the study was carried out. However, some of them did part of their practical training in other Spanish provinces or European countries thanks to student mobility programs. The same happened with students of the nursing degree. 

Learning should be a shared responsibility between students, and the academic and clinical institutions. As set out in the Standards and Guidelines for Quality Assurance in the European Higher Education Area [57], students also take on responsibilities in quality assurance related to learning, and they should be involved in the design and improvements of the study programmes. Therefore, they should speak out about their concerns about their learning-teaching process. However, the participants’ responsibility in their own learning process has not been explored, and their negative comments in the results section may lead to a biased view of the academic activity conducted by this university.

In order to change the curricula, it would be necessary to conduct further research throughout the upcoming years to confirm the obtained results. It would also be very useful to incorporate the lecturers’ visions of theory and clinical placements by means of multi-center studies to contrast opinions or carry out comparative studies that show whether or not such results are aligned.

## 5. Conclusions

Nursing care work varies in terms of the implementation of evidence-based care, with the coexistence of those nursing staff who rely on experience and those who rely on evidence to develop their activity.

EBP is promoted, but these findings suggest that it does not always transpire in practice, and a theory-practice gap has long been known to exist. Furthermore, despite the promotion of EBP, this research suggests that a theory-practice gap still exists, in relation to the lack of EBN, deficient coordination between the university and clinical institutions, lack of curricular orientation toward care practice, non-use of the NCP as a working method, certain forgetfulness of a holistic vision of the person, and sometimes, dehumanization of care once professional performance begins. This shows a complex scenario, and that it would require a great effort on stakeholders to work together on linking theory and practice in nursing.

At present, the conclusions suggest that these findings are not new, but that instead they lend support to a vast body of literature documenting similar findings.

## Figures and Tables

**Figure 1 ijerph-16-02774-f001:**
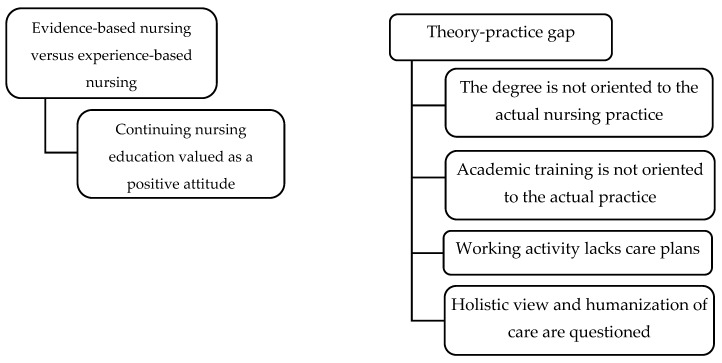
Themes and subthemes from the results. Source: prepared by the author.

**Table 1 ijerph-16-02774-t001:** The sociodemographic data of interviewees and participants of the focus groups.

Personal Interviews					Focus Groups		
Interview Code	Age	Sex	Work Experience in Months	Graduation Year	Focus Group Code	Age	Sex
E1	26	Woman	3	2017	GF1-P1	40	Woman
E2	23	Woman	4	2016	GF1-P2	21	Man
E3	25	Woman	7.5	2016	GF1-P3	21	Man
E4	23	Woman	3	2016	GF1-P4	23	Woman
E5	23	Man	3	2016	GF1-P5	22	Woman
E6	25	Woman	3	2015	GF1-P6	24	Woman
E7	23	Woman	9	2016	GF2-P7	21	Man
E8	28	Woman	6	2017	GF2-P8	23	Man
E9	25	Woman	6	2016	GF2-P9	51	Man
E10	28	Man	9	2017	GF2-P10	21	Man
E11	25	Man	12	2015	GF2-11	21	Woman
E12	51	Man	12	2015	GF2-P12	22	Man

Source: prepared by the author.

**Table 2 ijerph-16-02774-t002:** Primary questions of the interview guide.

Pre-Established Categories	Primary Questions
Clinical training	Which rotating internship was your favorite? Can you explain why?What nursing techniques did you practice? Could you practice them all?Were you involved in any difficult emotional situation that a patient/family member was going through? Can you explain how you felt and how you helped them?
Theoretical/Academic training	Are you satisfied with what you learned at a theoretical level during the degree? Can you explain your experience?Which courses do you think are most important? Can you explain why?Would you make any change to the academic curriculum?
Working situation	How do you evaluate your transition from study to work?What kind of difficulties have you experienced working as a nurse?Which circumstances cause you stress in your day-to-day work?

Source: prepared by the author.

**Table 3 ijerph-16-02774-t003:** Themes, subthemes, and codes.

Themes	Subthemes	Codes
Evidence-based nursing vs. experience-based nursing	Retrained vs. non-retrained nursing professionals	EBN in daily work, retrained nurses and stagnant nurses, no widespread use of guidelines and protocols, professionals working according to their experience, differences among EBN training professionals.
Theory-practice gap	The degree is not oriented to the actual nursing practice	Course not adapted to the actual healthcare situation, number of credits not adequate to the relevance of a course, reorganization of courses, contents not adapted to reality.
	Academic training not oriented to the actual practice	Adaptation of theory to actual healthcare situation, difference between theory and healthcare practice, reorganization of clinical placements, reorganization of courses.
	Working without care plans	Care plans not used in actual practice, too much attention to care plans in class.
	Holistic vision and humanization of care into question	Dehumanization at work, forgetting the holistic vision of the person, greater importance to techniques than to humanization and holistic vision.

Abbreviations: EBN, evidence-based nursing. Source: prepared by the author.

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
