# Peer review of "Assessment of Nursing Care and Teaching: A Qualitative Approach"

_ijerph, 2019, doi:10.3390/ijerph16152774_

Round 1

Reviewer 1 Report

Dear Authors

Thank you for this manuscript. I can say that I recognize your findings. However, I would like you to develop line 39 under introduction: what do you mean by this statement? how were the curricula designed that differed from before?

line 96, argue for why your inclusion criteria are sound. 

line 105: develop what you mean by data saturation was achieved, only the number of participants won't do as an argument.

line 111: what do you mean about triangulation? and how is it done in your paper? 

line 116: develop and clarify this part about collaborating students. how? why and in what way?

line 119: I do not understand what you mean by this statement, please clarify. 

line 129: develop on the field notes and how you used them.

I do lack a theoretical connection to learning theories, please add this and your result section will be more interesting.

I do want you to rewrite the result section adding more depth and not using so many citations. As it is now, it lacks clarity, depth, and stringency. It is to close to the raw data material, not jet analysed fully. What does it mean to be evidence-based in the different settings, how does it show, etc. 

I think this is your biggest challenge but please do add some depth so we as readers can understand the impact or thoughts of your result section. Is it hard to grasp the clinical setting if for instance correlation between the theoretical and practical worlds are lacking, is it harder to transition knowledge? 

After this is done rewrite the discussion part using the most valuable parts of the result section and connect it to the needs of patients and society. 

Thank you

Author Response

Cover letter – Reviewer 1.

Thank you very much for the suggested modifications for the paper Assessment of nursing care and teaching: A qualitative approach. Having them already done, all the authors believe that the article has improved.

First of all, we reconfirm the author list and the corresponding affiliation of them.

Every modification done is described below. In the paper you can find them in red colour:

- However, I would like you to develop line 39 under introduction: what do you mean by this statement? how were the curricula designed that differed from before?

We have added a sentence which sais that the Bologna Declaration unified nursing curricula and facilitated the validation of qualifications and mobility of nursing professionals throughout the European Union (line 36-38 in red)

- line 96, argue for why your inclusion criteria are sound.

We have added a sentence explaining that with these criteria, people who took part in the study had the necessary professional experience and recent training experience to provide data that responded to the research objectives (line 91-93 in red)

- line 105: develop what you mean by data saturation was achieved, only the number of participants won't do as an argument.

We have added a sentence to clarify it: “…as no data other than that already collected was provided since analysis categories were saturated” (line 100-101 in red)

- line 111: what do you mean about triangulation? and how is it done in your paper?

The sentence added (“The triangulation of data was intended to achieve reliability of the data obtained in personal interviews through a different source of information”) try to explain it (line 108-110 in red)

- line 116: develop and clarify this part about collaborating students. how? why and in what way?

We have clarify adding the sentence: “When asked about the quality of the teaching received, none of the groups was conducted by principal or collaborating researchers, with the aim of minimizing information bias between the students and the faculty who had given them both theoretical and practical lectures” (line 114-116 in red)

- line 119: I do not understand what you mean by this statement, please clarify.

No one else participated in the development of the focus groups.

This statement is in accordance to ítem 15 of COREQ criteria. If you consider this statement should be removed, we can do it (line 119 in red)

- line 129: develop on the field notes and how you used them.

We have added a sentence explaning the use of the field notes. (line 130-131 in red)

- I do lack a theoretical connection to learning theories, please add this and your result section will be more interesting.

We have added a mention to the social learning theory and its connection to variability in clinical practice, in the section Results (line 190-194 in red)

- I do want you to rewrite the result section adding more depth and not using so many citations. As it is now, it lacks clarity, depth, and stringency. It is to close to the raw data material, not jet analysed fully. What does it mean to be evidence-based in the different settings, how does it show, etc. I think this is your biggest challenge but please do add some depth so we as readers can understand the impact or thoughts of your result section. Is it hard to grasp the clinical setting if for instance correlation between the theoretical and practical worlds are lacking, is it harder to transition knowledge?

Results have been modified, removing citations and adding clarity, depth, and stringency (line 161-162, 173-175, 197-199, 205-206, 214-215, 223-225, 243-245, 258-260, 270-274, 281-282 in red)

After this is done rewrite the discussion part using the most valuable parts of the result section and connect it to the needs of patients and society.

Discussion has been rewrited including paragraphs according to modifications in the section results, and connecting to benefits for patients and society (line 331-336, 353-357, 361-367 in red)

Thank you. We hope to be hearing from you soon.

Reviewer 2 Report

Introduction – There is an abundance of existing research regarding the relationship between nursing education and practice, and more specifically, the nursing theory-practice gap.  Awareness of this research needs to be demonstrated, in order to contextualise the research, allow judgement as to its contribution to knowledge, and provide a rationale for the research.  The aim is presented rather abruptly, following introduction of contemporary developments in nursing education.  This transition could be smoothed by some mention of practice and existing literature, in order to contextualise and justify the research aim i.e. what informed this research aim/why was this aim considered to be important?  It could be made clearer in the background of the abstract and introduction that the aim of the research is informed by reform requirements for student participation in curriculum assessment (if indeed it is) i.e. the research was conducted as part of required local institutional evaluation.  And/or if the research was informed by a gap in the literature, this needs to be introduced and made clearer.  A wider project is mentioned but again, this does not contribute to providing a rationale.

Methods – The paper would benefit from the provision of a brief description of descriptive qualitative research as a methodology, derived from methodological literature, in order to contextualise the remainder of the paper and allow for judgement of quality, consistency and the appropriateness of such methodology.  The appropriateness of the methodology could also be more specifically related to the purpose underpinning the aim of the paper i.e. why this methodology was considered to be conducive to the purpose of the research.

Please include the number of participants invited in section 2.2, as it is stated that all invited participants participated.

12 personal interviews were conducted during an ‘authorized stay at the university’ but 3 were conducted via Skype, as they were not located at the university = 15 (or is this a reference?).  As written, it is unclear as to whether the 3 Skype interviews were in addition to the 12 personal interviews, or are Skype interviews included within the 12 personal interviews?  Similarly, regarding the focus groups ‘up to 12 people participated’ is a little confusing – how many people in total participated in the focus groups?  It also needs to be clearer as to whether the focus group participants consisted of participants who had participated in the Phase 1 interviews, or whether these were separate participants.  The demographic data collected for participants in the interviews and focus groups is also different – was there a reason for this?  It would have been interesting to see the work experience in months and graduation for focus group participants.  Is this data available for inclusion? ......Following writing these comments, I have deduced that focus group participants must have been students and personal interview participants qualified nurses (hence the reason for missing demographic data for focus group participants).  This needs to be explicitly identified in the paper.  It would also be beneficial to identify why both 4th year students and qualified nurses were invited as participants i.e. the contributions envisaged from each in terms of addressing the research aim.

Interview questions were informed by a literature review.  This increases the requirement for inclusion of such literature prior to this in the paper, as comment under ‘introduction’.

The interview questions would benefit from a supporting context, provided by a rationale for the aim of the research (as per comment under ‘introduction’), in order to ascertain their place and purpose within the research.

A brief description of Atlas.ti would be beneficial – what does this software do/achieve?  E.g. If it allows established codes to be applied to new data (as implied), via what means is this achieved – software finding keywords?

Results – ‘Two of the main themes that emerged…’ Were there other themes that are not reported in this paper (as this suggests), or does the paper report all themes (two)?  This needs to be clarified.

There is unreadable text in the box under ‘evidence-based nursing versus experience-based’ – only the top of the second line is shown.

3.1 Evidence-based nursing (and the place of protocols and clinical guidelines) needs to be introduced with the theory-practice gap literature, suggested under ‘introduction’.  Neither Evidence-Based Nursing or the theory-practice gap are introduced and described in the paper ahead of the findings.

3.1.1.  Unsure of the meaning of a ‘retrained’ and ‘non-retrained’ nurse – this needs to be introduced.  Whilst it is understood that this is a qualitative descriptive paper (therefore aimed at description and ‘close’ interpretation), there is very little analysis and interpretation presented in this section, which consists mainly of participant quotations, and the section is not related back to the research aim.

3.2.2.  Can the meaning of ‘courses’ be clarified in this section?  A course suggests a full nursing course.  Does this refer to a topic or module within a nursing course?

Discussion – the meaning of the first two sentences is not clear.  ‘This criticism’ – what criticism specifically?  In what way did they judge health care activity and training work?  Are these sentences intended to mean that participants were critical of healthcare activity and training work, and the insights that they provided are valuable for improvement activities?

Paragraphs 2 and 3 would read better as one paragraph.

‘reveal nurses’ hindrances to the use of this standardized language’ – the meaning of this is not entirely clear.  Language as a hindrance to nurses’ use of NCPs?

Limitations – could ‘in no case has the involvement…..’ be unpacked and explained briefly?

Conclusions – once background literature is added to the introduction, the conclusion would benefit from reference to what findings suggest in relation to the existing literature/knowledge (and by drawing on the narrative of the discussion).  This would assist in assessing the research’s contribution to existing knowledge.

Author Response

Cover letter – Reviewer 2

Comments to Suggestions

Thank you very much for the suggested modifications for the paper Assessment of nursing care and teaching: A qualitative approach. Having them already done, all the authors believe that the article has improved.

First of all, we reconfirm the author list and the corresponding affiliation of them.

Every modification done is described below in green colour. In the paper you can find them in blue colour:

Introduction – There is an abundance of existing research regarding the relationship between nursing education and practice, and more specifically, the nursing theory-practice gap.  Awareness of this research needs to be demonstrated, in order to contextualise the research, allow judgement as to its contribution to knowledge, and provide a rationale for the research.  The aim is presented rather abruptly, following introduction of contemporary developments in nursing education.  This transition could be smoothed by some mention of practice and existing literature, in order to contextualise and justify the research aim i.e. what informed this research aim/why was this aim considered to be important?  It could be made clearer in the background of the abstract and introduction that the aim of the research is informed by reform requirements for student participation in curriculum assessment (if indeed it is) i.e. the research was conducted as part of required local institutional evaluation.  And/or if the research was informed by a gap in the literature, this needs to be introduced and made clearer.  A wider project is mentioned but again, this does not contribute to providing a rationale.

Thank you for your valuable comments and suggestions for improving the Introduction section.

We have mention the nursing theory-practice gap and the evidence-based nursing practice in order to improve the contextualizations of the research. Besides the aim is now presented smoothly and it is clearer the goal of the main research and consequently, the Results reported in this paper. (line 47-48 and 53-61)

Methods – The paper would benefit from the provision of a brief description of descriptive qualitative research as a methodology, derived from methodological literature, in order to contextualise the remainder of the paper and allow for judgement of quality, consistency and the appropriateness of such methodology.  The appropriateness of the methodology could also be more specifically related to the purpose underpinning the aim of the paper i.e. why this methodology was considered to be conducive to the purpose of the research.

Thank you for your comment. A brief description of descriptive qualitative research has been added (line 68).

Please include the number of participants invited in section 2.2, as it is stated that all invited participants participated.

We have added the number of Participants invited as you required, also, we have clarified the number of the final Participants (line 84-88)

12 personal interviews were conducted during an ‘authorized stay at the university’ but 3 were conducted via Skype, as they were not located at the university = 15 (or is this a reference?).  As written, it is unclear as to whether the 3 Skype interviews were in addition to the 12 personal interviews, or are Skype interviews included within the 12 personal interviews?

We have rewrited the sentence to make it clearer (line 96-98)

Similarly, regarding the focus groups ‘up to 12 people participated’ is a little confusing – how many people in total participated in the focus groups?  It also needs to be clearer as to whether the focus group participants consisted of participants who had participated in the Phase 1 interviews, or whether these were separate participants.  The demographic data collected for participants in the interviews and focus groups is also different – was there a reason for this?  It would have been interesting to see the work experience in months and graduation for focus group participants.  Is this data available for inclusion? ......Following writing these comments, I have deduced that focus group participants must have been students and personal interview participants qualified nurses (hence the reason for missing demographic data for focus group participants).  This needs to be explicitly identified in the paper.  It would also be beneficial to identify why both 4th year students and qualified nurses were invited as participants i.e. the contributions envisaged from each in terms of addressing the research aim.

We have rewrited the paragraph and now is less confusing and you can clearly identify who participated in the focus group, the reason why and the data collected in table 1 (line 105-108, and 110-111)

Interview questions were informed by a literature review.  This increases the requirement for inclusion of such literature prior to this in the paper, as comment under ‘introduction’.

Thank you for your comment. The Introduction section have benn accordingly modified, but we are unsure if you request to add the bibliography we used to design the ad hoc questionnaire for  interviews and focus group. In any case here you have a table with this bibliography. If you wish, we can include it in the main text of the paper.

Bibliography in reverse order of publication year

-   Edward, KL., Ousey, K., Playle, J., Giandinoto, JA., 2017. Are new nurses   work ready – The impact of preceptorship. An integrative systematic review.   Journal of Professional Nursing 33, 326-333.

-   Meyer, G., Shatto, B., Delicath, T., Von der Lancken, S., 2017. Effect of   Curriculum Revision on Graduates’ Transition to Practice. Nurse Educator   42(3), 127-132.

- ReBueno, MCDR., Tiongco, DDD., Macindo, JRB., 2017. A structural equation model on the attributes of a skills enhancement   program affecting clinical competence of pre-graduate nursing students. Nurse   Education Today 49, 180-186.

- Beogo, I., Mendez Rojas, B., Gagnon, M.P., Liue,   C.Y. 2016. Psychometric evaluation of the French   version of the Clinical Nursing Competence Questionnaire (CNCQ-22): A   cross-sectional study in nursing education in Burkina Faso. Nurse Education Today 45,   173-178.

-   Wangensteen, S., Johansson, IS., Nordström, G., 2015. Nurse Competence Scale   – Psychometric testing in a Norwegian context. Nurse Education in Practice   15, 22-29.

-   European Federation of Nurses Association (EFN), 2015. Directriz de EFN para la   implementación del Artículo 31 de la Directiva 2005/36/CE sobre el   Reconocimiento Mutuo de Cualificaciones Profesionales, Enmendada por la   Directiva 2013/55/UE, en los programas nacionales de formación enfermera.

- Takase, M., Nakayoshi, Y.,   Teraoka, S., 2012. Graduate nurses’ perceptions of   mismatches between themselves and their jobs and association with intent to   leave employment: a longitudinal survey. International Journal of Nursing   Studies 49, 1521-1530.

- Boletín Oficial del Estado (BOE), 2008. Orden por la   que se establecen los requisitos para la verificación de los títulos   universitarios oficiales que habiliten para el ejercicio de la profesión de   Enfermero. Orden CIN/2134/2008 de 3 de julio. BOE 174, 31680-31683.

The interview questions would benefit from a supporting context, provided by a rationale for the aim of the research (as per comment under ‘introduction’), in order to ascertain their place and purpose within the research.

The Introduction sections has been modified according to your suggestions, and now the interview questions make more sense (line 47-48 and 53-61)

A brief description of Atlas.ti would be beneficial – what does this software do/achieve?  E.g. If it allows established codes to be applied to new data (as implied), via what means is this achieved – software finding keywords?

A brief dresciption of Atlas.ti has been added (line 143-144)

Results – ‘Two of the main themes that emerged…’ Were there other themes that are not reported in this paper (as this suggests), or does the paper report all themes (two)?  This needs to be clarified.

It is clarified that this paper is part of a bigger Research and only two themes are reported here (line 155-158)

There is unreadable text in the box under ‘evidence-based nursing versus experience-based’ – only the top of the second line is shown.

It has been modified and you can see clearer now (figure 1)

3.1 Evidence-based nursing (and the place of protocols and clinical guidelines) needs to be introduced with the theory-practice gap literature, suggested under ‘introduction’.  Neither Evidence-Based Nursing or the theory-practice gap are introduced and described in the paper ahead of the findings.

Thank you for your comment. This suggestion have also been done by the other reviewer. A sentence have been introduced to contextualize the topic (line 161-162 in red colour)

3.1.1.  Unsure of the meaning of a ‘retrained’ and ‘non-retrained’ nurse – this needs to be introduced.  Whilst it is understood that this is a qualitative descriptive paper (therefore aimed at description and ‘close’ interpretation), there is very little analysis and interpretation presented in this section, which consists mainly of participant quotations, and the section is not related back to the research aim.

We have changed the title of the subtheme and also we have used the term “continuing education” and “being up-to-date” to make clearer what we meant (line 179-182)

3.2.2.  Can the meaning of ‘courses’ be clarified in this section?  A course suggests a full nursing course.  Does this refer to a topic or module within a nursing course?

The meaning of courses have been clarified (line 223)

Discussion – the meaning of the first two sentences is not clear.  ‘This criticism’ – what criticism specifically?  In what way did they judge health care activity and training work?  Are these sentences intended to mean that participants were critical of healthcare activity and training work, and the insights that they provided are valuable for improvement activities?

We have rewrited the first paragraph in order to better explain the participants´ judgements on the topics addressed (line 287-288)

Paragraphs 2 and 3 would read better as one paragraph.

Done

‘reveal nurses’ hindrances to the use of this standardized language’ – the meaning of this is not entirely clear.  Language as a hindrance to nurses’ use of NCPs?

The meaning of the sentence is now clearer, after modifing the order of the words (line 322-323)

Limitations – could ‘in no case has the involvement…..’ be unpacked and explained briefly?

Limitations has been clarified and they are easier to understand now (line 399, 402-403)

Conclusions – once background literature is added to the introduction, the conclusion would benefit from reference to what findings suggest in relation to the existing literature/knowledge (and by drawing on the narrative of the discussion).  This would assist in assessing the research’s contribution to existing knowledge.

We have added a paragraph mentioning our contribution to existing knowledge (line 417-421)

Thank you. We hope to be hearing from you soon.

Round 2

Reviewer 1 Report

dear authors

I lack coherense and an elaborated ethics sextionde

Could you be clearer in your andalusiska section

Author Response

Cover letter – Reviewer 1.

Thank you very much for the suggested modification and for your time, for the paper Assessment of nursing care and teaching: A qualitative approach. Having it already done, all the authors believe that the article has improved.

The modification done is described below. In the paper you can find it in red colour:

- I lack coherense and an elaborated ethics sextionde. Could you be clearer in your andalusiska section.

We are sorry. We have interpreted your request as a need of making clearer if there is any specific Andalousian law in terms of ethical requirements for performing  researches. In Andalucía, just in case we had carried out our study in a clinical institution (involving staff and/or patients) it would had been mandatory to as for permission and ethical approval to the Portal de Ética de la Investigación Biomédica de Andalucía (Portal of Ethics of the Biomedical Research of Andalousia).

Our study involved students and newly-qualified nursing of a univiersity, therefore, the legal and ethical approval lie son the university.

We have make it clear in the ethical considerations section (line 152-156 in red colour).

Reviewer 2 Report

Cover letter – Reviewer 2

Comments to Suggestions

Thank you very much for the suggested modifications for the paper Assessment of nursing

care and teaching: A qualitative approach. Having them already done, all the authors

believe that the article has improved.

Many thanks for submitting this revised paper which is much improved.  The majority of comments made have been well addressed, however six comments require further addressing.  Please see responses below in green.

First of all, we reconfirm the author list and the corresponding affiliation of them.

Every modification done is described below in green colour. In the paper you can find them

in blue colour:

Introduction – There is an abundance of existing research regarding the relationship between

nursing education and practice, and more specifically, the nursing theory-practice

gap. Awareness of this research needs to be demonstrated, in order to contextualise the

research, allow judgement as to its contribution to knowledge, and provide a rationale for the

research. The aim is presented rather abruptly, following introduction of contemporary

developments in nursing education. This transition could be smoothed by some mention of

practice and existing literature, in order to contextualise and justify the research aim i.e. what

informed this research aim/why was this aim considered to be important? It could be made

clearer in the background of the abstract and introduction that the aim of the research is

informed by reform requirements for student participation in curriculum assessment (if

indeed it is) i.e. the research was conducted as part of required local institutional

evaluation. And/or if the research was informed by a gap in the literature, this needs to be

introduced and made clearer. A wider project is mentioned but again, this does not

contribute to providing a rationale.

Thank you for your valuable comments and suggestions for improving the Introduction

section.

We have mention the nursing theory-practice gap and the evidence-based nursing practice in

order to improve the contextualizations of the research. Besides the aim is now presented

smoothly and it is clearer the goal of the main research and consequently, the Results reported

in this paper. (line 47-48 and 53-61)

The introduction section is now much improved.  The sentence: ‘Evidence-based practice has become the standard of 55 clinical care [7], changing the traditional “experience-based nursing” model, thus renewing the 56 provision of services and guiding care to effectiveness, efficiency and patient safety. however, requires clarification.  EBP should but does not necessarily constitute the standard of clinical care – with the benefits indicated – hence the reason for the theory-practice gap.  i.e. although EBP is widely promoted, nurses do not consistently practice according to evidence (as you have found in this research), leading to a theory-practice gap.  This sentence needs to be clarified to ensure that it indicates that EBP is promoted, but not necessarily applied. 

‘In the 57 universities it is taught what nursing is, what are its competences and how to apply professional 58 care. This message should be in accordance with students’ experiences during their clinical practices 59 and the theory-practice gap might be minimized’.  The meaning of this requires clarification.  In saying that the message in universities should be in accordance with students’ experiences, is this suggesting that educators should ensure that what they teach reflects practice, or that practice reflects theory/evidence?

Overall, these sentences also require more referencing – particularly points made line 53, and 56.

Methods – The paper would benefit from the provision of a brief description of descriptive

qualitative research as a methodology, derived from methodological literature, in order to

contextualise the remainder of the paper and allow for judgement of quality, consistency and

the appropriateness of such methodology. The appropriateness of the methodology could

also be more specifically related to the purpose underpinning the aim of the paper i.e. why

this methodology was considered to be conducive to the purpose of the research.

Thank you for your comment. A brief description of descriptive qualitative research has been

added (line 68).

The comment here related to providing a brief description of what descriptive qualitative research is.  In addressing this comment, ‘descriptive qualitative research’ is stated but not described (line 68).

Please include the number of participants invited in section 2.2, as it is stated that all invited

participants participated.

We have added the number of Participants invited as you required, also, we have clarified

the number of the final Participants (line 84-88)

Addressed.

12 personal interviews were conducted during an ‘authorized stay at the university’ but 3

were conducted via Skype, as they were not located at the university = 15 (or is this a

reference?). As written, it is unclear as to whether the 3 Skype interviews were in addition

to the 12 personal interviews, or are Skype interviews included within the 12 personal

interviews?

We have rewrited the sentence to make it clearer (line 96-98)

Addressed.  Please add ‘with registered nurses’ to the Phase 1 narrative ‘semi-structured personal interviews were carried out [with registered nurses]’, so that it is clear on first reading that the interviews were with the registered nurses only and the focus groups were with the students.

Similarly, regarding the focus groups ‘up to 12 people participated’ is a little confusing –

how many people in total participated in the focus groups? It also needs to be clearer as to

whether the focus group participants consisted of participants who had participated in the

Phase 1 interviews, or whether these were separate participants. The demographic data

collected for participants in the interviews and focus groups is also different – was there a

reason for this? It would have been interesting to see the work experience in months and

graduation for focus group participants. Is this data available for inclusion? ......Following

writing these comments, I have deduced that focus group participants must have been

students and personal interview participants qualified nurses (hence the reason for missing

demographic data for focus group participants). This needs to be explicitly identified in the

paper. It would also be beneficial to identify why both 4th year students and qualified nurses

were invited as participants i.e. the contributions envisaged from each in terms of addressing

the research aim.

We have rewrited the paragraph and now is less confusing and you can clearly identify who

participated in the focus group, the reason why and the data collected in table 1 (line 105-

108, and 110-111)

This comment has been addressed well.

Interview questions were informed by a literature review. This increases the requirement for

inclusion of such literature prior to this in the paper, as comment under ‘introduction’.

Thank you for your comment. The Introduction section have benn accordingly modified, but

we are unsure if you request to add the bibliography we used to design the ad hoc

questionnaire for interviews and focus group. In any case here you have a table with this

bibliography. If you wish, we can include it in the main text of the paper.

Bibliography in reverse order of publication year

- Edward, KL., Ousey, K., Playle, J., Giandinoto, JA., 2017. Are new nurses work

ready – The impact of preceptorship. An integrative systematic review. Journal of

Professional Nursing 33, 326-333.

- Meyer, G., Shatto, B., Delicath, T., Von der Lancken, S., 2017. Effect

of Curriculum Revision on Graduates’ Transition to Practice. Nurse

Educator 42(3), 127-132.

- ReBueno, MCDR., Tiongco, DDD., Macindo, JRB., 2017. A structural equation

model on the attributes of a skills enhancement program affecting clinical competence

of pre-graduate nursing students. Nurse Education Today 49, 180-186.

- Beogo, I., Mendez Rojas, B., Gagnon, M.P., Liue, C.Y. 2016. Psychometric

evaluation of the French version of the Clinical Nursing Competence Questionnaire

(CNCQ-22): A cross-sectional study in nursing education in Burkina Faso. Nurse

Education Today 45, 173-178.

- Wangensteen, S., Johansson, IS., Nordström, G., 2015. Nurse Competence Scale –

Psychometric testing in a Norwegian context. Nurse Education in Practice 15, 22-29.

- European Federation of Nurses Association (EFN), 2015. Directriz de EFN para

la implementación del Artículo 31 de la Directiva 2005/36/CE sobre

el Reconocimiento Mutuo de Cualificaciones Profesionales, Enmendada por

la Directiva 2013/55/UE, en los programas nacionales de formación enfermera.

- Takase, M., Nakayoshi, Y., Teraoka, S., 2012. Graduate nurses’ perceptions

of mismatches between themselves and their jobs and association with intent

to leave employment: a longitudinal survey. International Journal of

Nursing Studies 49, 1521-1530.

- Boletín Oficial del Estado (BOE), 2008. Orden por la que se establecen los requisitos

para la verificación de los títulos universitarios oficiales que habiliten para el ejercicio

de la profesión de Enfermero. Orden CIN/2134/2008 de 3 de julio. BOE 174, 31680-

31683.

Many thanks for providing this bibliography.  Although I would suggest that the bibliography is not required in the paper, some of these papers could be used as references for the persistence of the theory-practice gap, despite recommendation and promotion of EBP, in the introduction (see comment regarding introduction).

The interview questions would benefit from a supporting context, provided by a rationale for

the aim of the research (as per comment under ‘introduction’), in order to ascertain their place

and purpose within the research.

The Introduction sections has been modified according to your suggestions, and now the

interview questions make more sense (line 47-48 and 53-61)

I agree, this has now been addressed.

A brief description of Atlas.ti would be beneficial – what does this software do/achieve? E.g.

If it allows established codes to be applied to new data (as implied), via what means is this

achieved – software finding keywords?

A brief dresciption of Atlas.ti has been added (line 143-144)

Thank you – this is now much clearer.

Results – ‘Two of the main themes that emerged…’ Were there other themes that are not

reported in this paper (as this suggests), or does the paper report all themes (two)? This

needs to be clarified.

It is clarified that this paper is part of a bigger Research and only two themes are reported

here (line 155-158)

Clarified well. 

There is unreadable text in the box under ‘evidence-based nursing versus experience-based’

– only the top of the second line is shown.

It has been modified and you can see clearer now (figure 1)

Addressed.

3.1 Evidence-based nursing (and the place of protocols and clinical guidelines) needs to be

introduced with the theory-practice gap literature, suggested under ‘introduction’. Neither

Evidence-Based Nursing or the theory-practice gap are introduced and described in the paper

ahead of the findings.

Thank you for your comment. This suggestion have also been done by the other reviewer. A

sentence have been introduced to contextualize the topic (line 161-162 in red colour)

Addressed here using this added sentence and inclusion of EBP in the introduction.  It is still required however that the new sentences regarding EBP and the theory-practice gap are clarified in the introduction (as per comment under ‘introduction’), to allow the sentence line 161 and 162 to make sense i.e. this sentence correctly states ‘should’, but the sentences in the introduction suggest that evidence is always applied in practice (see comment under ‘introduction’).

3.1.1. Unsure of the meaning of a ‘retrained’ and ‘non-retrained’ nurse – this needs to be

introduced. Whilst it is understood that this is a qualitative descriptive paper (therefore

aimed at description and ‘close’ interpretation), there is very little analysis and interpretation

presented in this section, which consists mainly of participant quotations, and the section is

not related back to the research aim.

We have changed the title of the subtheme and also we have used the term “continuing

education” and “being up-to-date” to make clearer what we meant (line 179-182)

Much improved and addressed.

3.2.2. Can the meaning of ‘courses’ be clarified in this section? A course suggests a full

nursing course. Does this refer to a topic or module within a nursing course?

The meaning of courses have been clarified (line 223)

Addressed.

Discussion – the meaning of the first two sentences is not clear. ‘This criticism’ – what

criticism specifically? In what way did they judge health care activity and training

work? Are these sentences intended to mean that participants were critical of healthcare

activity and training work, and the insights that they provided are valuable for improvement

activities?

We have rewrited the first paragraph in order to better explain the participants´ judgements

on the topics addressed (line 287-288)

Addressed.

Paragraphs 2 and 3 would read better as one paragraph.

Done

Addressed

‘reveal nurses’ hindrances to the use of this standardized language’ – the meaning of this is

not entirely clear. Language as a hindrance to nurses’ use of NCPs?

The meaning of the sentence is now clearer, after modifing the order of the words (line 322-

323)

Addressed.

Limitations – could ‘in no case has the involvement…..’ be unpacked and explained briefly?

Limitations has been clarified and they are easier to understand now (line 399, 402-403)

The intended meaning of this, and why it is a limitation is still not clear.  What does the participants’ responsibility in their own learning process mean, and why might this lead to a biased view of the academic activity conducted by the University (and in what way)?  This needs explaining further.

Conclusions – once background literature is added to the introduction, the conclusion would

benefit from reference to what findings suggest in relation to the existing

literature/knowledge (and by drawing on the narrative of the discussion). This would assist

in assessing the research’s contribution to existing knowledge.

We have added a paragraph mentioning our contribution to existing knowledge (line 417-

421)

The conclusions still require relating to existing literature (either in the discussion or conclusions)  i.e. that EBP is promoted but these findings suggest that it does not always transpire in practice, and a theory-practice gap has long been known to exist (references) and despite the promotion of EBP, this research suggests that a theory-practice gap still exists, in relation to (factors stated).  At present, the conclusions suggest that these findings are new – they are not new, but instead lend support to a vast body of literature documenting the similar findings.  This needs to be indicated.

Thank you. We hope to be hearing from you soon.

Author Response

Cover letter – Minor revisions Reviewer 2

Comments to Suggestions

Thank you very much for the suggested modifications for the paper Assessment of nursing care and teaching: A qualitative approach. Having them already done, all the authors believe that the article has improved.

Every modification done is described below in green colour. In the paper you can find them in green colour too:

Introduction - The introduction section is now much improved.  The sentence: ‘Evidence-based practice has become the standard of 55 clinical care [7], changing the traditional “experience-based nursing” model, thus renewing the 56 provision of services and guiding care to effectiveness, efficiency and patient safety. however, requires clarification.  EBP should but does not necessarily constitute the standard of clinical care – with the benefits indicated – hence the reason for the theory-practice gap.  i.e. although EBP is widely promoted, nurses do not consistently practice according to evidence (as you have found in this research), leading to a theory-practice gap.  This sentence needs to be clarified to ensure that it indicates that EBP is promoted, but not necessarily applied.

Thank you very much for your suggestion. We have added some sentences and their references. We think is much clearer (lines 53-61). 

‘In the 57 universities it is taught what nursing is, what are its competences and how to apply professional 58 care. This message should be in accordance with students’ experiences during their clinical practices 59 and the theory-practice gap might be minimized’.  The meaning of this requires clarification.  In saying that the message in universities should be in accordance with students’ experiences, is this suggesting that educators should ensure that what they teach reflects practice, or that practice reflects theory/evidence?

We think it has been clarified in lines 60-61 in text. 

Overall, these sentences also require more referencing – particularly points made line 53, and 56.

Added references number 7, 9, 19 and 11.

Methods - The comment here related to providing a brief description of what descriptive qualitative research is.  In addressing this comment, ‘descriptive qualitative research’ is stated but not described (line 68).

Thank you for your comment. We have added a sentence which describes what “descriptive qualitative research” is (lines 71-73).

Please add ‘with registered nurses’ to the Phase 1 narrative ‘semi-structured personal interviews were carried out [with registered nurses]’, so that it is clear on first reading that the interviews were with the registered nurses only and the focus groups were with the students.

Added (line 101).

Many thanks for providing this bibliography.  Although I would suggest that the bibliography is not required in the paper, some of these papers could be used as references for the persistence of the theory-practice gap, despite recommendation and promotion of EBP, in the introduction (see comment regarding introduction).

We have used references indicated in introduction section, because we think they are more appropriate.   

Results - Addressed here using this added sentence and inclusion of EBP in the introduction.  It is still required however that the new sentences regarding EBP and the theory-practice gap are clarified in the introduction (as per comment under ‘introduction’), to allow the sentence line 161 and 162 to make sense i.e. this sentence correctly states ‘should’, but the sentences in the introduction suggest that evidence is always applied in practice (see comment under ‘introduction’).

Thank you for your comments for results section. As we explain in comments for the introduction section, we have clarified that EBN practice is not always adhered to. 

Limitations – The intended meaning of this, and why it is a limitation is still not clear.  What does the participants’ responsibility in their own learning process mean, and why might this lead to a biased view of the academic activity conducted by the University (and in what way)?  This needs explaining further.

Thank you for your suggestion in limitations subheading. We have added a paragraph (lines 410-417), explaining what does the participants responsibility in their own learning process means. So we have added a reference for sound it.

Conclusions – The conclusions still require relating to existing literature (either in the discussion or conclusions)  i.e. that EBP is promoted but these findings suggest that it does not always transpire in practice, and a theory-practice gap has long been known to exist (references) and despite the promotion of EBP, this research suggests that a theory-practice gap still exists, in relation to (factors stated).  At present, the conclusions suggest that these findings are new – they are not new, but instead lend support to a vast body of literature documenting the similar findings.  This needs to be indicated.

Thank you for your comment in conclusions section. We have modified the conclusions sections using your suggestions. We think that now is more according to results and limitations (lines 426-435).

Thank you. We hope to be hearing from you soon.

Round 3

Reviewer 2 Report

I can confirm that the authors have successfully addressed all comments. They have inserted new text into the manuscript in green, which has some English language/grammatical etc errors. The only information I couldn't amend is line 72, as I was unsure as to the meaning of this. Information in brackets indicates the English language changes required. Line 57: which foment[s] Line 61: That [what] is taught in class Line 71: aim[s] Line 72: Setting use and relationship between categories. (I'm not sure what is meant here). Line 416: negative comments on Results section [negative comments in the Results section]